# Shining light on vitamin C deficiency and scurvy in Canada: A scoping review protocol of risk profiles, health outcomes, and interventions

Chiemena Ndukwe[1]☸, Nnamdi Ndubuka[1,2,3]*, Justina Ndubuka[1,3]☸, Carol Udey[3]¤, Emmanuel Dankwah[1,3]☸

**1** School of Public Health, University of Saskatchewan, Saskatoon, Saskatchewan, Canada,
**2** Department of Community Health and Epidemiology, University of Saskatchewan, Saskatoon, Saskatchewan, Canada, **3** Northern Inter-Tribal Health Authority, Prince Albert, Saskatchewan, Canada

☸ These authors contributed equally to this work.
¤ Current Address: Community Service Unit, Northern Inter-Tribal Health Authority, Prince Albert, Saskatchewan, Canada
* nnamdi.ndubuka@usask.ca

## Abstract

Scurvy, resulting from vitamin C deficiency, is linked to serious health outcomes, including impaired collagen synthesis, anemia, and delayed wound healing. Once considered largely eradicated in high-income countries, scurvy has re-emerged among specific Canadian populations, driven by factors such as inadequate dietary intake, socioeconomic disparities, and limited access to nutritious foods. Despite growing awareness, evidence regarding its prevalence, risk factors, and public health responses in Canada remains sparse and fragmented. To address this knowledge gap, this study protocol outlines a scoping review designed to: (a) assess the prevalence and incidence of scurvy and vitamin C deficiency in Canada; (b) identify at-risk populations and contributing factors; (c) describe associated health outcomes; and (d) map existing nutritional interventions and public health strategies for prevention and management. Our review will follow the Joanna Briggs Institute (JBI) Manual for Evidence Synthesis (Chapter 11: Scoping review) and the Arksey and O'Malley methodological framework. Keywords will be identified and used to develop search strings used for a comprehensive search of various databases. The review will adhere to the Preferred Reporting Items for Systematic Reviews and Meta-Analyses extension for Scoping Reviews(PRISMA-ScR) guidelines. Our study selection process will systematically screen titles/abstracts and full texts of potentially relevant articles to ensure a comprehensive analysis through thematic analysis. We will implement a clearly defined extraction process to gather the most pertinent articles, maximizing the quality and impact of our research. Ethical approval is not required because this study will review publicly available data and will not involve human participants. Our scoping review will synthesize evidence on scurvy and vitamin C deficiency in Canada, identifying knowledge gaps, contributing factors such as food insecurity, and vulnerable

**Data availability statement:** No datasets were generated or analysed during the current study. All relevant data from this study will be made available upon study completion.

**Funding:** The author(s) received no specific funding for this work.

**Competing interests:** The authors have declared that no competing interests exist.

populations. Findings will inform research priorities, guide public health policies, and support targeted interventions to prevent deficiency and enhance nutritional health for Canadians.

## Introduction

Scurvy was first observed in 1550 BC among soldiers and sailors who had little or no access to fresh fruits and vegetables [1]. Scurvy, as a disease, is caused by vitamin C deficiency in the body [2]. Vitamin C, being an essential nutrient, is required by the body for collagen production, iron absorption, and immunological response, and is also an important physiological antioxidant [3,4]. The human body's inability to synthesize vitamin C, combined with the important role played by vitamin C in immune development and total body functioning, makes the body dependent on exogenous sources for vitamin C intake [4]. Although scurvy is considered rare, evidence suggests that the rare nature of the disease might be attributed to underdiagnosis or underreporting due to its nonspecific symptoms and the lack of awareness among healthcare professionals [5].

Several factors have been suggested to predispose populations to scurvy, including food insecurity, limited access to fresh fruits, income status, social lifestyles, dietary habit changes, and the impacts of colonization [1,2,6]. Fatigue, perifollicular hemorrhages, bleeding gums, bone fragility, epiphyseal fractures, subperiosteal hemorrhage, bone pain, difficulty walking, and poor wound healing are some symptoms of vitamin C deficiency [1,7,8].

Scurvy was first observed in Canada between 1535–1536 among Jacques Cartier's crew in Quebec [9]. Although much decreased in prevalence, Scurvy still exists today in Canada, with a recent resurgence of cases in Northern Saskatchewan (La Ronge) [10]. In the past, Indigenous peoples, such as the Inuit, maintained healthy vitamin C levels by consuming traditional diets rich in raw fish, organ meats, and other ascorbic acid-rich animal products [11]. However, these Indigenous populations are today more predisposed to scurvy than the non-indigenous populations, which has been demonstrated to be associated withfood insecurity, income status, social lifestyles, greater reliance on low-nutrient market foods, breakdown of traditional agricultural systems, climate change, and nutrition transition [12]. Following the quest to combat the resurgence of scurvy and vitamin C deficiency in Canada, examining the prevalence, trends, circumstances of outbreaks, and health system responses towards diagnosis and treatment of scurvy and vitamin C deficiency becomes crucial.

This scoping review is therefore necessary for identifying, mapping, and synthesizing available information regarding scurvy and vitamin C deficiency among Indigenous and non-indigenous populations in Canada. The scoping review aims to capture the prevalence and incidence of scurvy and vitamin C deficiency, its predisposing factors, health system responses, public health implications, and existing literature gaps. The review will also be instrumental in developing key messages and sourcing evidence, which will be vital in informing policy changes geared towards combating scurvy and vitamin C deficiency in Canada.

Specifically, this scoping review will address the following questions:

a. What risk factors are associated with vitamin C deficiency and scurvy in Canadian populations?

b. What are the clinical manifestations and health outcomes linked to vitamin C deficiency and scurvy in the Canadian context?

c. What clinical management practices, public health strategies, and nutrition programs have been implemented in Canada to prevent vitamin C deficiency and scurvy?

d. What gaps exist in the current literature, policies, or interventions related to vitamin C deficiency and scurvy in Canada?

## Materials and methods

### Study design

A scoping review will be conducted following the Joanna Briggs Institute (JBI) Manual for Evidence Synthesis (Chapter 11: Scoping Reviews) [13] and the Arksey and O'Malley methodological framework [14], which provides a systematic approach to mapping key concepts, evidence types, and knowledge gaps in emerging or heterogeneous fields. In addition, the review will be guided by the Preferred Reporting Items for Systematic Reviews and Meta-Analyses extension for Scoping Reviews (PRISMA-ScR) guidelines [15] to ensure methodological transparency and reproducibility. The study protocol is registered in the Open Science Framework (OSF), Registration DOI: https://doi.org/10.17605/OSF.IO/U7HPK [16].

We aim to comprehensively identify, map, and synthesize evidence on scurvy and vitamin C deficiency among Canadian populations, including both Indigenous and non-Indigenous groups. Specifically, the review will address the prevalence and incidence of scurvy and vitamin C deficiency, predisposing factors, health outcomes, health system responses, public health implications, and existing gaps in literature and policy.

### Search strategy

We will develop a robust search strategy in consultation with an experienced librarian at the University of Saskatchewan. Keywords and controlled vocabulary (e.g., MeSH terms) will be identified and combined into database-specific search strings using Boolean operators and proximity connectors. Keywords will cover the main concepts of scurvy, vitamin C deficiency, risk profiles, health outcomes, interventions, Canadian context, and the context of geographic location (Table 1). All factors suggested to predispose indigenous and non-indigenous populations to scurvy and vitamin C deficiency, as mentioned in the introduction section, are included under the broader category of risk, risk profiles, risk factors, predisposing factors, and determinants. Synonyms, alternate spellings, and related terms will be included to maximize comprehensiveness.

We will conduct searches in multiple electronic databases, including Medline, PubMed, Embase, CINAHL, Global Health, Google Scholar, Scopus, Web of Science, Statistics Canada, and the Public Health Database. Each database will be searched using tailored syntax and controlled vocabulary to ensure sensitivity and specificity. We will review literature published from January 2000 to December 2025 to capture historical context, changes in dietary patterns, contemporary food insecurity issues, diagnostic practices, and updated public health policies. We specifically selected the year 2000 as the starting point to include the pre-re-emergence period and provide a comprehensive understanding of trends in scurvy and vitamin C deficiency in the Canadian population.

All retrieved references will be uploaded to Mendeley for reference management and secure backup. The references will then be exported to Covidence, an online systematic review management platform, to facilitate de-duplication, screening, data extraction, and collaboration among reviewers (https://app.covidence.org). The reference lists of included studies

**Table 1. List of keywords, synonyms, and MeSH terms.**

| Keywords | Synonyms | MeSH Terms |
|---|---|---|
| **Scurvy** | Vitamin C deficiency<br>Ascorbic acid deficiency<br>Moeller–Barlow disease<br>Nutritional scurvy<br>Micronutrient deficiency | exp Scurvy/<br>exp Ascorbic Acid Deficiency/ |
| **Vitamin C deficiency** | Hypovitaminosis C<br>Ascorbate deficiency<br>Ascorbic acid deficiency<br>Low Vitamin C | exp Ascorbic Acid Deficiency/ |
| **Risk** | Risk factors<br>Predisposing factors<br>Determinants<br>Sociodemographic factors | exp Risk Factors/ |
| **Health outcomes** | Clinical manifestations<br>Symptoms<br>Signs<br>Health effects<br>Morbidity<br>Mortality | |
| **Interventions** | Nutritional supplementation<br>Food fortification<br>Dietary programs<br>Public health programs<br>Treatment<br>Nutritional education | |
| **Canada** | Canadian*<br>Northern America<br>Geospatial<br>Geographic<br>Regional<br>Neighborhood<br>Rural<br>Remote<br>Northern<br>Urban<br>Geodemographic<br>Spatial analysis | exp Canada/ |

will be screened to identify additional relevant literature. Institutional repositories and grey literature sources from recognized health organizations will also be explored to ensure comprehensive coverage.

## Inclusion and exclusion criteria

Population: We will include studies that report on scurvy or vitamin C deficiency in the Canadian population. Studies covering individuals of all ages and genders, including Indigenous and non-Indigenous populations, will be considered.

Concept: The review will encompass literature addressing the prevalence, incidence, predisposing factors, clinical manifestations, health outcomes, nutritional interventions, public health strategies, and policy responses related to scurvy and vitamin C deficiency. Studies discussing gaps and future recommendations will also be included.

Context: Recent reports suggest that social, cultural, and economic factors may contribute to the re-emergence of scurvy in Canada. The review will examine these factors to provide a comprehensive understanding of the determinants and public health implications of vitamin C deficiency. The review will also examine reports or studies that utilized

geospatial science and tools such as geospatial informatics systems (GIS) and geodemographic segmentation method-ology to study the incidence/prevalence of scurvy and vitamin C deficiency, and predisposing factors like income status, social lifestyles, and dietary habits. Our inclusion of the context of geographic location is because several researchers have, over time, pointed to a regional association with scurvy and an increased risk of insufficient vitamin C intake.

We will include a broad range of sources, including primary research articles, systematic reviews, meta-analyses, case reports/case series, policy papers, conference proceedings, websites, and grey literature. No restrictions will be applied to study design, ensuring a comprehensive mapping of available evidence. Only literature published in English will be considered.

We will exclude studies that are duplicates, preprints, editorials, opinion pieces, news articles. Studies conducted out-side Canada or unrelated to scurvy or vitamin C deficiency will be excluded. Biochemical studies without human popula-tion data, non-English literature, conference abstracts without full texts, and non-human studies will also be excluded.

## Study selection process

All sourced literature from databases will be uploaded to Mendeley and subsequently exported to Covidence. Covidence will automatically remove duplicate entries; any remaining duplicates will be manually excluded by reviewers during screening. Screening will occur in three stages: (a) title and abstract screening, (b) full-text screening, and (c) reference list and citation screening of included studies.

Two independent reviewers (CN and JN) will conduct title and abstract screening, blinded to each other's decisions. Articles that meet all inclusion criteria will progress to full-text review. Full texts will be independently assessed by the same reviewers, and inclusion will require consensus. Discrepancies will be resolved through discussion or by consult-ing a third reviewer (NN). This multi-stage, independent review process minimizes bias and ensures reliability. The study selection process will be illustrated using a PRISMA-ScR flow diagram.

## Data extraction

We will use the predefined keywords in our search strategy to identify studies that match the review objectives in the data extraction. We will design a standardized data extraction form based on these key concepts to capture relevant variables and outcomes consistently. This alignment ensures methodological coherence, minimizes selection and information bias, and strengthens the transparency, rigor, and reproducibility of the review. The form will be piloted on a subset of studies and revised as needed to ensure consistency and comprehensiveness. The standardized data extraction form will include the following variables: author(s), year of publication, study location, study design, population characteristics, prev-alence and incidence of scurvy or vitamin C deficiency, identified risk factors, health outcomes, interventions, public health strategies, and gaps in knowledge or policy. Public health strategies hereinafter refer to initiatives or interventions that have been adopted historically or currently to address scurvy and vitamin C deficiency among the Canadian population, including population-level programs, guidelines, educational campaigns, nutritional interventions, and policy measures. The data extraction will be completed immediately after the full-text screening of available literature.

The primary reviewer (CN) will extract data, with secondary reviewers (JN, NN, ED, CU) providing oversight and valida-tion. Discrepancies will be resolved through discussion to achieve consensus.

## Data analysis and synthesis

Qualitative content analysis will be used to synthesize extracted data. We will adopt Braun and Clarke's inductive-deductive thematic approach [17] to identify recurring patterns, trends, and gaps across the literature. Using Braun and Clarke's inductive-deductive thematic approach, we will group the extracted data into themes such as epidemiology (prev-alence, incidence, risk profiles), health outcomes, and interventions or public health strategies, as well as highlight gaps

in evidence and policy. Findings will be presented using narrative summaries, tables, flow diagrams, and evidence maps to provide a clear visual representation of the scope, distribution, and nature of the evidence. The evidence maps will be created as matrices that show themes across key factors like population groups (age, indigenous identity, socioeconomic status), settings (community, clinical, institutional), and regions within Canada. These evidence maps will help show where evidence is strong and where gaps exist. Tables will include detailed summaries of study characteristics (author, year, location, design, population, outcomes, interventions), as well as summary tables that group findings by themes, such as types of risk factors, health outcomes, and interventions or policy categories. These tables will also note where evidence is limited. Flow diagrams will show the study selection process and, when relevant, how codes develop into themes.

We will also provide narrative summaries with all visual materials to explain how the evidence we map and summarize addresses the review's goal and research questions. We will look for geographic patterns and point out where geospatial methods or region-specific analyses are missing. The review will provide a comprehensive overview of scurvy and vitamin C deficiency in Canada, highlighting at-risk populations, social determinants of health, public health responses, and policy implications. Evidence gaps identified through this synthesis will inform future research priorities and guide the development of targeted public health interventions.

## Ethical considerations and declarations

Our scoping review exclusively involves the synthesis of publicly available literature and does not include human participants or identifiable personal data; therefore, formal ethical approval is not required. The study will be conducted following established best practices for transparency, rigor, and responsible research conduct. All data and extracted information will be securely stored using password-protected platforms, including Mendeley and Covidence, ensuring integrity and confidentiality of the reviewed sources.

## Discussion

This scoping review aims to systematically synthesize existing evidence on scurvy and vitamin C deficiency in Canada, focusing on prevalence, incidence, risk factors, health outcomes, and interventions. By examining both Indigenous and non-Indigenous populations, the review will highlight social determinants of health, including food insecurity, socio-economic disparities, and dietary transitions, which may contribute to the re-emergence of this preventable condition [10,12]. Understanding these factors is critical for informing evidence-based public health policies and targeted nutritional interventions.

Several limitations are inherent to the study design. First, restricting inclusion to English-language publications may exclude relevant non-English literature. Second, reliance on publicly available and grey literature may result in publication bias, as community-level or unpublished data could be missed. Third, scoping reviews do not assess causality, limiting conclusions about intervention effectiveness [13,14]. Additionally, heterogeneity in study designs and reporting practices may challenge synthesis and interpretation.

Dissemination plans include publication in peer-reviewed journals, conference presentations, and distribution through professional networks. Summaries of findings will be shared with policymakers, public health practitioners, and community organizations to support evidence-informed interventions and policies.

Any amendments to the study protocol, such as changes in search strategy, inclusion criteria, or data extraction methods, will be documented transparently. Should termination of the review be necessary due to unforeseen challenges, this will be reported alongside a summary of completed work and rationale.

Overall, this review will provide a comprehensive mapping of evidence on scurvy and vitamin C deficiency in Canada, identify gaps in knowledge and policy, and guide future research priorities. By synthesizing available evidence and evaluating public health responses, the study aims to support targeted interventions and policies to prevent vitamin C deficiency and improve nutritional health among Canadian populations.

## Supporting information

**S1 Appendix. Eligibility criteria for literature screening.**
(DOCX)

**S1 File. PRISMA-P 2015 checklist.**
(DOCX)

## Acknowledgments

We would like to acknowledge and thank Jessi Robinson, a Librarian at the University of Saskatchewan, for her support during the development of the search strategy used for our database search.

## Author contributions

**Conceptualization:** Nnamdi Ndubuka.

**Formal analysis:** Emmanuel Dankwah.

**Investigation:** Nnamdi Ndubuka, Emmanuel Dankwah.

**Methodology:** Chiemena Ndukwe, Nnamdi Ndubuka, Justina Ndubuka, Emmanuel Dankwah.

**Project administration:** Nnamdi Ndubuka, Justina Ndubuka.

**Resources:** Nnamdi Ndubuka, Carol Udey.

**Software:** Chiemena Ndukwe.

**Supervision:** Nnamdi Ndubuka, Justina Ndubuka.

**Validation:** Justina Ndubuka, Carol Udey.

**Visualization:** Nnamdi Ndubuka.

**Writing – original draft:** Chiemena Ndukwe, Nnamdi Ndubuka.

**Writing – review & editing:** Chiemena Ndukwe, Nnamdi Ndubuka, Justina Ndubuka, Carol Udey, Emmanuel Dankwah.

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
