## [Decision Letter · Decision Letter 0]

18 Dec 2025

Dear Dr. Ndubuka,

Thank you for submitting your manuscript to PLOS ONE. After careful consideration, we feel that it has merit but does not fully meet PLOS ONE’s publication criteria as it currently stands. Therefore, we invite you to submit a revised version of the manuscript that addresses the points raised during the review process.

We look forward to receiving your revised manuscript.

Kind regards,

Subburaman Mohan

Academic Editor

PLOS One

Reviewers' comments:

Reviewer's Responses to Questions

**Comments to the Author**

1. Does the manuscript provide a valid rationale for the proposed study, with clearly identified and justified research questions?

Reviewer #1: Yes

Reviewer #2: Yes

2. Is the protocol technically sound and planned in a manner that will lead to a meaningful outcome and allow testing the stated hypotheses?

Reviewer #1: Yes

Reviewer #2: Partly

3. Is the methodology feasible and described in sufficient detail to allow the work to be replicable?

Reviewer #1: Yes

Reviewer #2: Yes

4. Have the authors described where all data underlying the findings will be made available when the study is complete?

Reviewer #1: Yes

Reviewer #2: Yes

5. Is the manuscript presented in an intelligible fashion and written in standard English?

Reviewer #1: Yes

Reviewer #2: Yes

You may also provide optional suggestions and comments to authors that they might find helpful in planning their study.

Reviewer #1: This is a well-structured and clearly written study protocol for a scoping review. The research question is relevant and timely, the methodology is sound and follows established frameworks.

However, there are several areas where the manuscript can be improved to enhance its clarity, rigor, and impact.

Under-Specified Data Charting (Extraction) Process: The data extraction section is too vague. A scoping review's credibility hinges on a transparent and reproducible data charting process. The manuscript mentions a "standardized data extraction form" but does not detail the specific variables to be extracted. For example, what “public health strategies” mean. This should be explicitly listed and explained.

What is the relation between the key words in the searching strategy and the data extraction?

The description of how the data will be synthesized and presented is minimal. The plan to use "narrative summaries, tables, flow diagrams, and evidence maps" is good, but it needs more detail. How will the evidence be mapped? What will the tables contain? A clearer plan is needed.

The choice of literature from "January 2000 to August 2025" is arbitrary without a stated rationale. Why 2000? Is this to capture the "re-emergence" in a modern context? This justification should be explicitly provided.

The decision to exclude case reports is a significant limitation, especially for a re-emerging condition like scurvy in a high-income country. Early evidence and clinical notes often appear first in case reports. This exclusion should be strongly justified, or the criteria should be reconsidered.

Minor Issues

Authors stated that “Data extract will be completed in November 2025” And “Final results will be expected in January 2026”. Does that mean the work has been done before this protocol being published?

Reviewer #2: This review by Ndukwe et al. addresses the prevalence and incidence of vitamin C deficiency a, predisposing factors, health outcomes, health system responses and public health implications among Canadian populations. Scurvy, resulting from vitamin C deficiency is linked to serious health outcomes and continues to be present at high rates among certain populations. While this is an important topic, there are some issues in the review raised below that need the author’s attention during the revision.

General Comments

1) Please provide more details regarding the reason(s) why the stated scoping review methodology was chosen for this work.

2) The protocol has not included the context of geographic location in the search strategy. Numerous reports are emerging that demonstrate the utility of geospatial science and tools such as geospatial informatics systems (GIS) and geodemographic segmentation methodology to study incidence, prevalence, income status, social lifestyles, and dietary habits. Reference #9 (Dangerfield K.) cited by the authors directly points to a regional association with scurvy and an increased risk of insufficient vitamin C intake by residents of Northern Saskatchewan (La Ronge). In addition, the authors cite “limited access to fruit” as a factor associated with vitamin C deficiency. This also likely has a location-based feature. Please modify the manuscript to include the context of location. For example, in Table 1, under the Canada category, add terms related geospatial science such as geospatial, geographic, location, region, and neighborhood. Because the literature on scurvy is sparse, it is likely these terms will not return results. However, an important knowledge gap may be identified.

3) Lines 91-93 state that Indigenous populations are predisposed to scurvy “due to food insecurity, income status, social lifestyles, greater reliance on low nutrient market foods, breakdown of traditional agricultural systems, climate change, and nutrition transition [11].” There are several issues with this passage.

o First, the words “due to” imply causal relationships have been established for each of these factors. Due to the observational nature of several studies cited, many of these factors have only an established association (e.g., food insecurity and low vitamin C) not a causal relationship (e.g., vitamin C deficiency causes scurvy). This issue is also present in Line 79, where it is stated “have been proven”. Consider “suggest” or “have been demonstrated to be associated with”. Please review the entire manuscript for other places where authors may have erroneously implied a causal relationship exists and revise the text.

o A second issue is that the passage includes potential risk factors that are not directly addressed by the study search methodology. I am in agreement that there likely are associations between low vitamin C intake and “the breakdown of traditional agricultural systems, climate change, and nutrition transition” (i.e., reduced consumption of traditional diet elements). Additionally, Line 81 mentions other Indigenous population-relevant factors: “dietary change” and “colonization”. However, it is unclear to me how these factors which are specific to Indigenous populations will be captured by the search terms outlined in Table 1.

4) As described in Gandhi (cited reference #1), smoking and alcoholism are risk factors for vitamin C deficiency. And although the literature is sparse, studies have reported higher prevalence rates of these factors in Indigenous populations. These potential factors should be addressed in the manuscript.

Specific comments

• In the Title capitalize “light”

• Abtract Lines 39-40 “PRISMAScR (Preferred Reporting Items for Systematic Reviews and Meta-Analyses extension for Scoping Reviews)” reverse what is in parentheses

• Abstract Lines 52-53: Consider changing “Risk profile” to “Risk” as a Key Word to make your article more discoverable. Also, intervention should be capitalized for consistency.

• Reference 2 “Persistent scurvy after vitamin C supplementation in a high-risk patient: a case report” does not seem to be the best reference for the mechanism of actions related to hypovitaminosis C.

• Table 1:

o Under “Vitamin C deficiency” category, remove the word “level” behind “low vitamin C”.

o In addition to “symptoms” (i.e. subjective observations) also Include “signs” (i.e., objective clinical observations) under the “Clinical Outcomes” category

• Under the “Canada” category, it is unclear what the asterisk in “Canadian*” is notating.

• Line 81 Needs a reference regarding “colonization” as a risk factor

• Line 101:“Predisposing” should not be capitalized

• Lines 101-103: “scurvy and vitamin C deficiency” is repeated multiple times. Please revise the redundancy. Consider using a colon and reorganizing the sentence

• Line 123 typo “Inaddition”

• Line 130: change “of vitamin” to “of scurvy and vitamin”

• Line 205 change “November” to “December”

• Line 232 - C “deficiency in Canada, focusing on prevalence”, this sentence should include “incidence”

**Do you want your identity to be public for this peer review?** For information about this choice, including consent withdrawal, please see our Privacy Policy

Reviewer #1: **Yes:** Weikuan Gu

Reviewer #2: No

---

## [Author Response · Author response to Decision Letter 1]

15 Feb 2026

February 15th, 2026

Re: Title: Shining light on Vitamin C Deficiency and Scurvy in Canada: A Scoping Review Protocol of Risk Profiles, Health Outcomes, and Interventions. [PONE-D-25-56435]

Dear Editor,

Thank you for providing your valuable feedback as well as that of the reviewers. We have provided point-by-point responses and addressed all the comments. Please see the authors’ responses highlighted in the relevant sections below.

Response to Editorial Comments

Thank you for submitting your manuscript to PLOS ONE. After careful consideration, we feel that it has merit but does not fully meet PLOS ONE’s publication criteria as it currently stands. Therefore, we invite you to submit a revised version of the manuscript that addresses the points raised during the review process.

Response: We sincerely appreciate your kind consideration of our manuscript as having merit and for the constructive feedback provided by the reviewers. We also appreciate all the time and effort invested in this review process, all aimed at ensuring this manuscript fully meets PLOS ONE’s publication criteria. We carefully acknowledged each point raised during the review process and have thoroughly addressed them to ensure that the revised version of the manuscript fully meets PLOS ONE’s publication criteria.

Response: Thank you for approving our request to extend the resubmission deadline to February 15, 2026.

• A letter that responds to each point raised by the academic editor and reviewer(s). You should upload this letter as a separate file labeled 'Response to Reviewers'.

Response: We acknowledge the items listed to be submitted with the revised manuscript. We thoroughly drafted a letter that responds to the points raised by the academic editor and reviewer(s) and labelled it ‘Response to Reviewers' as directed. We also prepared a marked-up copy of our manuscript, which highlighted all changes made to the original version, and we uploaded it as a separate file labelled ‘Revised Manuscript with Track Changes’ as instructed. In the same vein, we uploaded a separate unmarked copy of our manuscript without tracked changes and labeled it ‘Manuscript’ as directed.

Response: We appreciate your clarification concerning the financial disclosure statement. We affirm that our financial disclosure will not be revised, and the statement stays as it was initially submitted.

If applicable, we recommend that you deposit your laboratory protocols in protocols.io to enhance the reproducibility of your results. Protocols.io assigns your protocol its own identifier (DOI) so that it can be cited independently in the future. Additionally, PLOS ONE offers an option for publishing peer-reviewed Lab Protocol articles, which describe protocols hosted on protocols.io.

Response: We appreciate the recommendation about depositing laboratory protocols to improve reproducibility and thank you for your guidance on this matter. At this point, we would like to carefully state that our study protocol has already been registered with the Open Science Framework (OSF), making it publicly available and time-stamped to promote transparency and reproducibility. Currently, we have not submitted the protocol to protocols.io; however, we recognize the advantages of this platform and value the information regarding both protocol deposition and the option to publish a peer-reviewed Lab Protocol article.

We are also open to exploring the possibility of linking our existing OSF registration with protocols.io to synchronize our protocol across both platforms, thereby further enhancing accessibility and the reproducibility of our protocol.

Response: We appreciate you bringing this to our attention. We thoroughly examined these requirements and confirmed that the updated version of the manuscript adheres to all of PLOS ONE’s formatting guidelines. Additionally, we ensured that the files submitted with the manuscript were appropriately labeled according to the instructions.

When completing the data availability statement of the submission form, you indicated that you will make your data available on acceptance. We strongly recommend all authors decide on a data sharing plan before acceptance, as the process can be lengthy and hold up publication timelines. Please note that, though access restrictions are acceptable now, your entire data will need to be made freely accessible if your manuscript is accepted for publication. This policy applies to all data except where public deposition would breach compliance with the protocol approved by your research ethics board. If you are unable to adhere to our open data policy, please kindly revise your statement to explain your reasoning and we will seek the editor's input on an exemption. Please be assured that, once you have provided your new statement, the assessment of your exemption will not hold up the peer review process.

Response: Thank you for the valuable comment. We are confirming that the entire data underlying our research will be freely available and accessible, without restriction

PLOS requires an ORCID iD for the corresponding author in Editorial Manager on papers submitted after December 6th, 2016. Please ensure that you have an ORCID iD and that it is validated in Editorial Manager. To do this, go to ‘Update my Information’ (in the upper left-hand corner of the main menu), and click on the Fetch/Validate link next to the ORCID field. This will take you to the ORCID site and allow you to create a new iD or authenticate a pre-existing iD in Editorial Manager.

Response: Thank you for the comment. The corresponding author has an ORCID iD and this has been validated in the PLOSOne Editorial Manager.

Response: Thank you for this clarification. We carefully reviewed the reviewer comments and confirmed that none of the reviewers recommended citing specific previously published works. We are also not aware of any prior publications within our context that directly relate to this manuscript and warrant citation. Therefore, we did not add any additional references.

Response to Reviewers Comments

Reviewer 1

This is a well-structured and clearly written study protocol for a scoping review. The research question is relevant and timely, the methodology is sound and follows established frameworks. However, there are several areas where the manuscript can be improved to enhance its clarity, rigor, and impact.

Response: We sincerely thank the reviewer for this thoughtful and encouraging comment. We appreciate your recognition of the relevance and timeliness of our research question and your acknowledgement of our methodology, which follows established frameworks. Your feedback has been invaluable, and we have carefully considered your suggestions to enhance further the relevance, impact, and clarity of our work.

Under-Specified Data Charting (Extraction) Process: The data extraction section is too vague. A scoping review's credibility hinges on a transparent and reproducible data charting process. The manuscript mentions a "standardized data extraction form" but does not detail the specific variables to be extracted. For example, what “public health strategies” mean. This should be explicitly listed and explained.

Response: We thank the reviewer for this insightful comment and for highlighting the importance of a transparent and reproducible data charting process in scoping reviews. We have clarified that our standardized data extraction form will include the following variables: author name(s), year of publication, study location, study design, population characteristics, prevalence/incidence of scurvy or vitamin C deficiency, identified risk factors, health outcomes, interventions, public health strategies, and gaps in knowledge or policy. We have also explicitly explained “public health strategies” as initiatives or interventions that have been adopted historically or currently to address scurvy and vitamin C deficiency among the Canadian population, including population-level programs, guidelines, educational campaigns, nutritional interventions, and policy measures.

What is the relation between the key words in the searching strategy and the data extraction?

Response: Thank you for this important comment. We have revised the first paragraph of the data extraction section to clearly describe the linkage between the search strategy and data extraction. In the manuscript, we now state: “We will use the predefined keywords in our search strategy to identify studies that match the review objectives in the data extraction. We will design a standardized data extraction form based on these key concepts to capture relevant variables and outcomes consistently. This alignment ensures methodological coherence, minimizes selection and information bias, and strengthens the transparency, rigor, and reproducibility of the review.”

The description of how the data will be synthesized and presented is minimal. The plan to use "narrative summaries, tables, flow diagrams, and evidence maps" is good, but it needs more detail. How will the evidence be mapped? What will the tables contain? A clearer plan is needed.

Response: We thank the reviewer for this insightful comment and for pointing out the need to clarify how we will synthesize and present our results. We have revised this section to explain that, using Braun and Clarke’s inductive and deductive thematic approach, we will group the extracted data into themes such as epidemiology (prevalence, incidence, risk profiles), health outcomes, and interventions or public health strategies, as well as highlight gaps in evidence and policy. We will create evidence maps as matrices that show these themes across key factors like population groups (age, Indigenous identity, socioeconomic status), settings (community, clinical, institutional), and regions within Canada. This will help show where evidence is strong and where gaps exist. Our tables will include detailed summaries of study characteristics (author, year, location, design, population, outcomes, and interventions), as well as summary tables that group findings by theme, such as types of risk factors, health outcomes, and intervention or policy categories. These tables will also note where evidence is limited. Flow diagrams will show the study selection process and, when relevant, how codes develop into themes. We will also provide narrative summaries with all visual materials to explain how the evidence we map and summarize addresses the review’s goals and research questions.

The choice of literature from "January 2000 to August 2025" is arbitrary without a stated rationale. Why 2000? Is this to capture the "re-emergence" in a modern context? This justification should be explicitly provided.

Response: We thank the reviewer for this important comment. Reports of scurvy re-emergence in modern developed countries began appearing in the medical literature around 2007. We selected the year 2000 as the starting point to capture the pre-re-emergence period, allowing us to contextualize trends and better understand the evolution of scurvy and vitamin C deficiency in the Canadian population.

The decision to exclude case reports is a significant limitation, especially for a re-emerging condition like scurvy in a high-income country. Early evidence and clinical notes often appear first in case reports. This exclusion should be strongly justified, or the criteria should be reconsidered.

Response: Thank you for your thoughtful comment. We have considered the reviewer’s suggestion and have included both case reports and case series to ensure we capture early evidence and clinical observations in case reports. This revision strengthens the comprehensiveness and relevance of our review.

Authors stated that “Data extract will be completed in November 2025” And “Final results will be expected in January 2026”. Does that mean the work has been done before this protocol being published?

Response: Thank you for this comment. These were projected timelines for our project with the assumption that our protocol will be published within a short timeframe. After reviewing all comments by reviewers, we have adjusted our work plan accordingly. We will commence literature search once our protocol is approved for publication.

Reviewer 2

This review by Ndukwe et al. addresses the prevalence and incidence of vitamin C deficiency a, predisposing factors, health outcomes, health system responses and public health implications among Canadian populations. Scurvy, resulting from vitamin C deficiency is linked to serious health outcomes and continues to be present at high rates among certain populations. While this is an important topic, there are some issues in the review raised below that need the author’s attention during the revision.

Response: Thank you for your helpful feedback on our review and for pointing out how important this topic is for Canadian populations. We appreciate the chance to revise and improve the manuscript, and we have addressed all the issues you raised.

Please provide more details regarding the reason(s) why the stated scoping review methodology was chosen for this work.

Response: Thank you for your comment. We followed the JBI Manual and the Arksey and O’Malley framework to provide a clear and organized way to map key concepts, evidence types, and knowledge gaps. We also used PRISMA ScR to make sure our methods and reporting are rigorous and reproducible. We chose our search strategy and evidence sources to fit the goals of a scoping review. By searching multiple databases, grey literature, and institutional repositories, we aimed to find sparse and fragmented evidence, since scurvy is often underdiagnosed and underreported worldwide. We included all study designs, systematic reviews, policy papers, conference proceedings, websites, and grey literature, using a Population–Concept–Context framework focused on Canada. This approach will help us map not only epidemiologic patterns but also clinical management, nutrition programs, and policy responses.

We chose our study selection, data extraction, and synthesis methods to ensure a systematic and reliable review of diverse evidence. Two independent reviewers screened studies in three stages, with a third reviewer resolving any disagreements, to reduce bias and improve consistency. We used Mendeley and Covidence to remove duplicates, track decisions, and document the process clearly with a PRISMA-ScR flow diagram. The adoption of a standardized extraction form will help us collect comparable data on populations, study designs, prevalence or incidence, risk factors, health outcomes, interventions, and policy gaps from different sources. For analysis, we adopted Braun and Clarke’s inductive and deductive thematic approach to ensure qualitative content analysis, which will help us find patterns, trends, and gaps instead of pooled effect estimates, since the studies were so varied. We chose to present our findings with narrative summaries, tables, flow diagrams, and evide

---

## [Editor Report · Decision Letter 1]

24 Feb 2026

Shining light on Vitamin C Deficiency and Scurvy in Canada: A Scoping Review Protocol of Risk Profiles, Health Outcomes, and Interventions.

PONE-D-25-56435R1

Dear Dr. Ndubuka,

We’re pleased to inform you that your manuscript has been judged scientifically suitable for publication and will be formally accepted for publication once it meets all outstanding technical requirements.

Kind regards,

Subburaman Mohan

Academic Editor

PLOS One
---

## [Editor Report · Acceptance letter]

PONE-D-25-56435R1

PLOS One

Dear Dr. Ndubuka,

I'm pleased to inform you that your manuscript has been deemed suitable for publication in PLOS One. Congratulations! Your manuscript is now being handed over to our production team.

Kind regards,

on behalf of

Dr. Subburaman Mohan

Academic Editor

PLOS One